# TaxiBGC: a Taxonomy-Guided Approach for Profiling Experimentally Characterized Microbial Biosynthetic Gene Clusters and Secondary Metabolite Production Potential in Metagenomes

Vinod K. Gupta,[a,b] Utpal Bakshi,[c] Daniel Chang,[d] Aileen R. Lee,[e,l] John M. Davis III,[f] Sriram Chandrasekaran,[g,h,i] Yong-Su Jin,[j,k] Michael F. Freeman,[e,l] Jaeyun Sung[a,b,f]

[a]Microbiome Program, Center for Individualized Medicine, Mayo Clinic, Rochester, Minnesota, USA

[b]Division of Surgery Research, Department of Surgery, Mayo Clinic, Rochester, Minnesota, USA

[c]Institute of Health Sciences, Presidency University, Kolkata, West Bengal, India

[d]Department of Computer Science and Engineering, University of Minnesota, Minneapolis, Minnesota, USA

[e]Department of Biochemistry, Molecular Biology, and Biophysics, University of Minnesota—Twin Cities, St. Paul, Minnesota, USA

[f]Division of Rheumatology, Department of Medicine, Mayo Clinic, Rochester, Minnesota, USA

[g]Department of Biomedical Engineering, University of Michigan, Ann Arbor, Michigan, USA

[h]Program in Chemical Biology, University of Michigan, Ann Arbor, Michigan, USA

[i]Center for Bioinformatics and Computational Medicine, University of Michigan, Ann Arbor, Michigan, USA

[j]Carl R. Woese Institute for Genomic Biology, University of Illinois at Urbana-Champaign, Urbana, Illinois, USA

[k]Department of Food Science and Human Nutrition, University of Illinois at Urbana-Champaign, Urbana, Illinois, USA

[l]BioTechnology Institute, University of Minnesota—Twin Cities, St. Paul, Minnesota, USA

**ABSTRACT** Biosynthetic gene clusters (BGCs) in microbial genomes encode bioactive secondary metabolites (SMs), which can play important roles in microbe-microbe and host-microbe interactions. Given the biological significance of SMs and the current profound interest in the metabolic functions of microbiomes, the unbiased identification of BGCs from high-throughput metagenomic data could offer novel insights into the complex chemical ecology of microbial communities. Currently available tools for predicting BGCs from shotgun metagenomes have several limitations, including the need for computationally demanding read assembly, predicting a narrow breadth of BGC classes, and not providing the SM product. To overcome these limitations, we developed taxonomy-guided identification of biosynthetic gene clusters (TaxiBGC), a command-line tool for predicting experimentally characterized BGCs (and inferring their known SMs) in metagenomes by first pinpointing the microbial species likely to harbor them. We benchmarked TaxiBGC on various simulated metagenomes, showing that our taxonomy-guided approach could predict BGCs with much-improved performance (mean $F_1$ score, 0.56; mean PPV score, 0.80) compared with directly identifying BGCs by mapping sequencing reads onto the BGC genes (mean $F_1$ score, 0.49; mean PPV score, 0.41). Next, by applying TaxiBGC on 2,650 metagenomes from the Human Microbiome Project and various case-control gut microbiome studies, we were able to associate BGCs (and their SMs) with different human body sites and with multiple diseases, including Crohn's disease and liver cirrhosis. In all, TaxiBGC provides an *in silico* platform to predict experimentally characterized BGCs and their SM production potential in metagenomic data while demonstrating important advantages over existing techniques.

**IMPORTANCE** Currently available bioinformatics tools to identify BGCs from metagenomic sequencing data are limited in their predictive capability or ease of use to even computationally oriented researchers. We present an automated computational pipeline called TaxiBGC, which predicts experimentally characterized BGCs (and infers

Address correspondence to Jaeyun Sung, Sung.Jaeyun@mayo.edu.

The authors declare no conflict of interest.

their known SMs) in shotgun metagenomes by first considering the microbial species source. Through rigorous benchmarking techniques on simulated metagenomes, we show that TaxiBGC provides a significant advantage over existing methods. When demonstrating TaxiBGC on thousands of human microbiome samples, we associate BGCs encoding bacteriocins with different human body sites and diseases, thereby elucidating a possible novel role of this antibiotic class in maintaining the stability of microbial ecosystems throughout the human body. Furthermore, we report for the first time gut microbial BGC associations shared among multiple pathologies. Ultimately, we expect our tool to facilitate future investigations into the chemical ecology of microbial communities across diverse niches and pathologies.

**KEYWORDS** biosynthetic gene cluster, natural product, secondary metabolite, metagenomics, microbiome, biomarkers, bacteriocin

Microbial secondary metabolites (SMs) (also called natural products) are a group of low-molecular-weight, structurally diverse, and bioactive chemical compounds (1–3). Although not essential for the growth, development, and reproduction of microorganisms, these compounds—which include bacterial and fungal cell-to-cell signaling molecules, pigments, bacteriocins, and siderophores—are known to primarily mediate key interactions within microbial ecosystems (4–7). Interestingly, these microbial SMs have recently been found to substantially regulate many aspects of human health and disease (8–12). Furthermore, microbial SMs are a significant source of antibiotics, antifungals, anticancer agents, immunosuppressants, and other pharmaceutical drugs (13–18). For example, alkaloids were found to have cytotoxic activity against human cancer cell lines (13) and to inhibit biofilm formation of methicillin-resistant *Staphylococcus aureus* (MRSA) and vancomycin-resistant *Enterococci* (VRE) (19).

The genes mainly responsible for synthesizing, modifying, and exporting SMs in microbes are often found in biosynthetic gene clusters (BGCs). In brief, BGCs are physically clustered groups of genes in a microbial genome that encode a biosynthetic pathway for producing an SM and its chemical variants (20, 21). With the advent of large-scale genomic studies and the ensuing big data revolution in natural products research (22), researchers have been able to apply computational genome mining to discover novel microbial BGCs and SMs (1, 10, 20, 23), including those from human microbiomes (16, 24). Hence, bioinformatics methods hold strong promise for detecting BGCs and inferring their SM products.

Early computational tools to detect BGCs from microbial genomes (e.g., antiSMASH [25], CLUSEAN [26], and PRISM [27]) use heuristic, rule-based algorithms based on the compositional similarity of gene/protein domains to annotated reference BGCs. Other tools that utilize machine learning models have lately emerged, demonstrating a greater ability to discover novel BGCs. One such widely used method is ClusterFinder (21), which employs hidden Markov models for BGC detection. Additionally, DeepBGC (28) implements recurrent neural networks for BGC identification. Moreover, Navarro-Muñoz et al. (29) have provided an integrated computational workflow consisting of two software tools (BiG-SCAPE and CORASON) for identifying novel gene cluster families and their phylogenetic relationships from microbial genomes. Furthermore, the primary specialized metabolic gene clusters (MGCs) of anaerobic bacteria were studied extensively using gutSMASH (30); this Web server determines the metabolic potential of anaerobic bacteria by predicting both known and putative MGCs using the same Pfam domain detection rules used by antiSMASH. Notably, gutSMASH is specifically designed to identify MGCs from the gut microbiome. Lastly, Pascal Andreu et al. (31) have recently introduced BiG-MAP, an automated command-line tool for profiling the abundance and expression of gene clusters in metagenomic and metatranscriptomic samples, respectively.

Approaches to predict BGCs have expanded beyond assembled or complete genomes to enable BGC prediction from culture-independent microbial communities using shotgun metagenomic sequencing. One such method is BiosyntheticSPAdes (32), which utilizes assembly graphs to detect clusters encoding for nonribosomal peptide synthetases (NRPSs)

and polyketide synthases (PKSs) from genomic and metagenomics data sets. In addition, MetaBGC (24) demonstrated the characterization of BGCs directly from metagenomic reads. This method first detects BGC reads from shotgun metagenome sequences based on sequence-scoring models; afterward, the identified reads are binned for targeted assembly to reconstruct novel BGCs.

Despite the variety of approaches to detect BGCs from microbial genomes and metagenomes, currently available tools have several shortcomings. Approaches such as antiSMASH (25), CLUSEAN (26), and PRISM (27) are designed to predict BGCs of known pathway classes from only individual microbial genomes. Machine learning-based methods can predict unknown BGCs, but these methods can generate more false-positive predictions than rule-based approaches (33). Moreover, these rule-based and machine learning-based approaches require assembled genome (or metagenome) sequences to detect BGCs, limiting the scalability of their application toward metagenomic data sets. Among BGC detection strategies for metagenomes, BiosyntheticSPAdes is limited by the requirement of read assembly before BGC prediction and thus requires considerable computational time and memory consumption. MetaBGC, which is an assembly independent method, can be used directly on metagenomic reads for BGC prediction; however, as mentioned in its original publication (24), the requirement of laborious parameter optimization and the unavailability of prebuilt models for a BGC of interest limit the utility of this tool for a wide range of metagenomic data sets. Furthermore, neither of these two metagenome-based BGC detection methods pinpoint the specific microbes that harbor the predicted BGC genes, leaving researchers uncertain as to where the BGCs could be coming from. Clearly, there is a need to improve current computational tools to enable accurate, rapid, and unbiased predictions of BGCs—and their corresponding SMs—from shotgun metagenomes of microbial communities.

Realizing the limitations mentioned above, we set out to develop a method that accurately and rapidly detects known BGCs from real-world, complex microbiomes. Herein, we introduce taxonomy-guided identification of biosynthetic gene clusters (TaxiBGC), an easy-to-use, command-line tool that identifies experimentally characterized BGCs from shotgun metagenomic data and infers their known SM products. TaxiBGC does not require genome assembly and can be applied to any metagenome. Importantly, our novel approach enables the prediction of experimentally characterized BGCs by first considering the microbial species from which they are derived, thereby allowing the user to trace the predicted and likely taxonomic origins of BGCs.

## RESULTS

**TaxiBGC for profiling experimentally characterized biosynthetic gene clusters in metagenomes.** TaxiBGC (Taxonomy-guided Identification of Biosynthetic Gene Clusters) is a computational strategy for predicting experimentally characterized BGCs and their annotated SMs in shotgun metagenomic sequencing data. The TaxiBGC pipeline consists of three major steps (Fig. 1A). The first step of the TaxiBGC pipeline performs species-level, taxonomic profiling on the metagenome using MetaPhlAn3 (34). The second step performs the first-pass prediction of BGCs through querying these species (identified in the first step) in the TaxiBGC reference database (Fig. 1B)—a predefined collection of 390 unique species with their experimentally characterized BGCs and known SMs (Table 1; see Table S1 in the supplemental material). The last step of the TaxiBGC pipeline performs *in silico* confirmation of the predicted BGCs (from the second step) based on read mapping (i.e., alignment) using BBMap (35). Here, mapping of shotgun sequencing reads onto BGC genes is conducted with a predetermined pair of minimum BGC gene presence and BGC coverage thresholds. Finally, the SMs corresponding to the confirmed BGCs are retrieved from the TaxiBGC reference database. TaxiBGC is completely open access for anyone to apply to their own metagenomic data set (see "Data Availability" for links to the TaxiBGC GitHub repository and conda installation).

In our study, the presence of a BGC gene was determined based on the fulfillment of a minimum percentage length of that gene mapped onto by metagenomic reads; and the coverage of a BGC was defined as the proportion of its total number of genes that were found to be present in the metagenome. We determined a set of

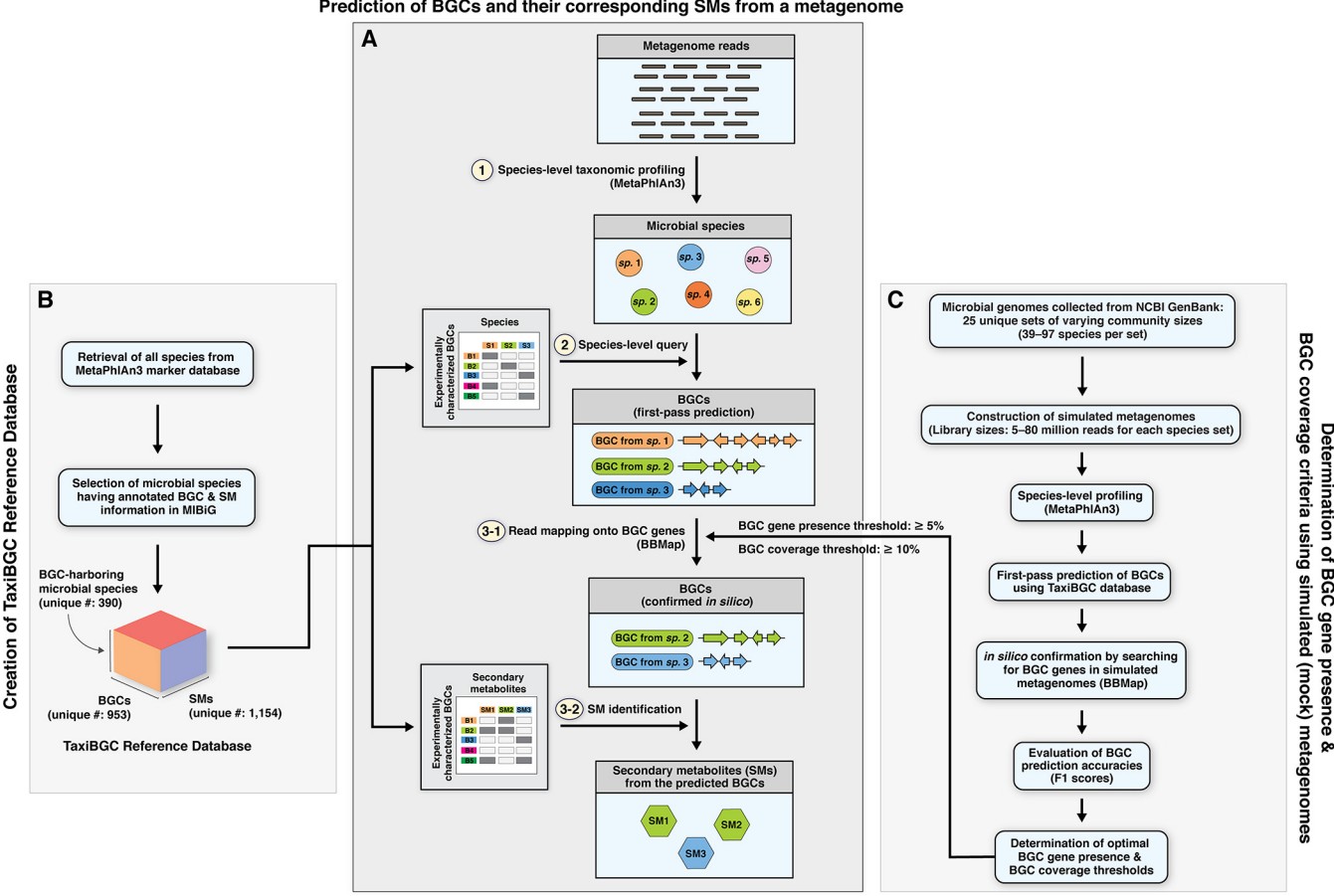

**FIG 1** Schematic overview of the TaxiBGC pipeline. (A) The prediction of BGCs and their corresponding SMs from a microbiome (i.e., metagenome) sample can be summarized in three main steps: first, species-level taxonomic profiling is performed using MetaPhlAn3; second, all identified species are queried in the TaxiBGC reference database (see B) as the first-pass prediction of which experimentally characterized BGCs may be present in the sample; and third, as an *in silico* confirmation of the predicted BGCs, metagenomic reads are mapped (aligned) onto those BGCs using BBMap with a set of predetermined minimum BGC gene presence and BGC coverage thresholds (see C). For the BGCs whose presence is confirmed, their corresponding SM products are retrieved from the TaxiBGC reference database. (B) The TaxiBGC reference database provides annotated information on experimentally characterized BGCs and their corresponding SMs. To construct the TaxiBGC reference database, BGC and SM information was retrieved from the MIBiG database (as of January 2022). The TaxiBGC reference database encompasses 390 BGC-harboring microbial species, 953 experimentally characterized BGCs, and 1,154 SMs. (C) The BGC gene presence and BGC coverage thresholds of TaxiBGC were determined based on prediction accuracies on simulated metagenomes of mock microbial communities. Simulated metagenomes were created from 25 unique sets of various microbial community sizes (39 to 97 unique species) at 5 different library sizes (5 million, 10 million, 20 million, 40 million, and 80 million paired-end reads) (see Materials and Methods). Afterward, as described in A, BGCs were predicted in all 125 simulated metagenomes to find the optimal pair of thresholds for the TaxiBGC pipeline. Here, the BGC gene presence is defined as the percentage length of a BGC gene covered by metagenomic reads; and BGC coverage is defined as the proportion of the total number of genes of BGC that are mapped onto by metagenomic reads. Prediction accuracy in the form of $F_1$ scores can be obtained by comparing the predicted and actual BGCs comprising the simulated metagenomes of known compositions. A pair of "BGC gene presence: 5%" and "BGC coverage: 10%" thresholds resulted in the highest mean $F_1$ score across a total of 125 simulated metagenomes and were thus selected as the optimal (default) thresholds used in A.

optimal BGC gene presence and BGC coverage thresholds for the TaxiBGC pipeline based on their overall predictive performance on 125 simulated metagenomes of mock microbial communities (Fig. 1C). These simulated metagenomes were constructed using 25 unique sets of various microbial community sizes and library sizes (Materials and Methods). More specifically, analogous to the steps described in Fig. 1A, we predicted BGCs in simulated metagenomes accordingly, as follows: (i) species-level taxonomic profiling, (ii) initial prediction of experimentally characterized BGCs, and (iii) *in silico* confirmation of BGC predictions wherein BGC genes were searched for in the metagenomes (via read mapping) using a range of BGC gene presence and BGC coverage thresholds. We obtained prediction accuracy ($F_1$ scores) by comparing the predicted BGCs with the actual BGCs comprising the simulated metagenomes. The pair of thresholds yielding the highest mean $F_1$ score after testing

**TABLE 1** Taxonomic summary of the BGC-harboring microbial species in the TaxiBGC reference database

| Kingdom | Phylum | Class | No. of species | No. of strains[a] | No. of unique BGCs | No. of unique BGCs by BGC class[b] | | | | | | | No. of unique SMs[b] |
|---|---|---|---|---|---|---|---|---|---|---|---|---|---|
| | | | | | | Alkaloid | NRP | Polyketide | RiPP | Saccharide | Terpene | Other | |
| Archaea | Euryarchaeota | Methanomicrobia | 1 | 1 | 1 | 0 | 0 | 0 | 0 | 0 | 0 | 1 | 1 |
| Bacteria | Actinobacteria | Actinobacteria | 163 | 209 | 393 | 8 | 92 | 158 | 59 | 45 | 27 | 54 | 515 |
| | Bacteroidetes | Bacteroidia | 3 | 6 | 8 | 0 | 0 | 0 | 0 | 4 | 0 | 4 | 8 |
| | | Chitinophagia | 2 | 2 | 3 | 0 | 0 | 1 | 1 | 0 | 0 | 1 | 3 |
| | | Flavobacteria | 3 | 3 | 3 | 0 | 1 | 1 | 0 | 0 | 0 | 1 | 3 |
| | Chloroflexi | Chloroflexia | 1 | 1 | 2 | 0 | 1 | 0 | 0 | 0 | 1 | 0 | 2 |
| | Cyanobacteria | Cyanobacteria | 18 | 32 | 42 | 4 | 20 | 21 | 10 | 1 | 1 | 1 | 56 |
| | Firmicutes | Bacilli | 54 | 71 | 128 | 0 | 20 | 5 | 77 | 22 | 1 | 6 | 129 |
| | | Clostridia | 8 | 8 | 9 | 0 | 0 | 0 | 8 | 0 | 0 | 1 | 8 |
| | Proteobacteria | Alphaproteobacteria | 19 | 20 | 23 | 0 | 3 | 2 | 6 | 4 | 4 | 5 | 37 |
| | | Betaproteobacteria | 20 | 26 | 43 | 1 | 30 | 19 | 3 | 4 | 0 | 2 | 52 |
| | | Deltaproteobacteria | 14 | 20 | 65 | 0 | 48 | 59 | 1 | 0 | 2 | 1 | 82 |
| | | Epsilonproteobacteria | 2 | 2 | 2 | 0 | 1 | 1 | 0 | 1 | 0 | 0 | 2 |
| | | Gammaproteobacteria | 65 | 81 | 149 | 0 | 52 | 25 | 16 | 44 | 2 | 26 | 161 |
| Eukaryote | Ascomycota | Eurotiomycetes | 17 | 24 | 82 | 2 | 29 | 46 | 2 | 0 | 8 | 7 | 110 |
| Total | | | 390 | 506 | 953 | 15 | 297 | 338 | 183 | 125 | 46 | 110 | 1,169[c] |

[a]Strain information from MIBiG (January 2022).
[b]According to MIBiG (January 2022).
[c]There are a total of 1,169 SMs linked to all strains in the TaxiBGC reference database. However, since strains of different taxonomic classes can have BGCs that encode the same SM, the number of unique SMs in the reference database is 1,154.

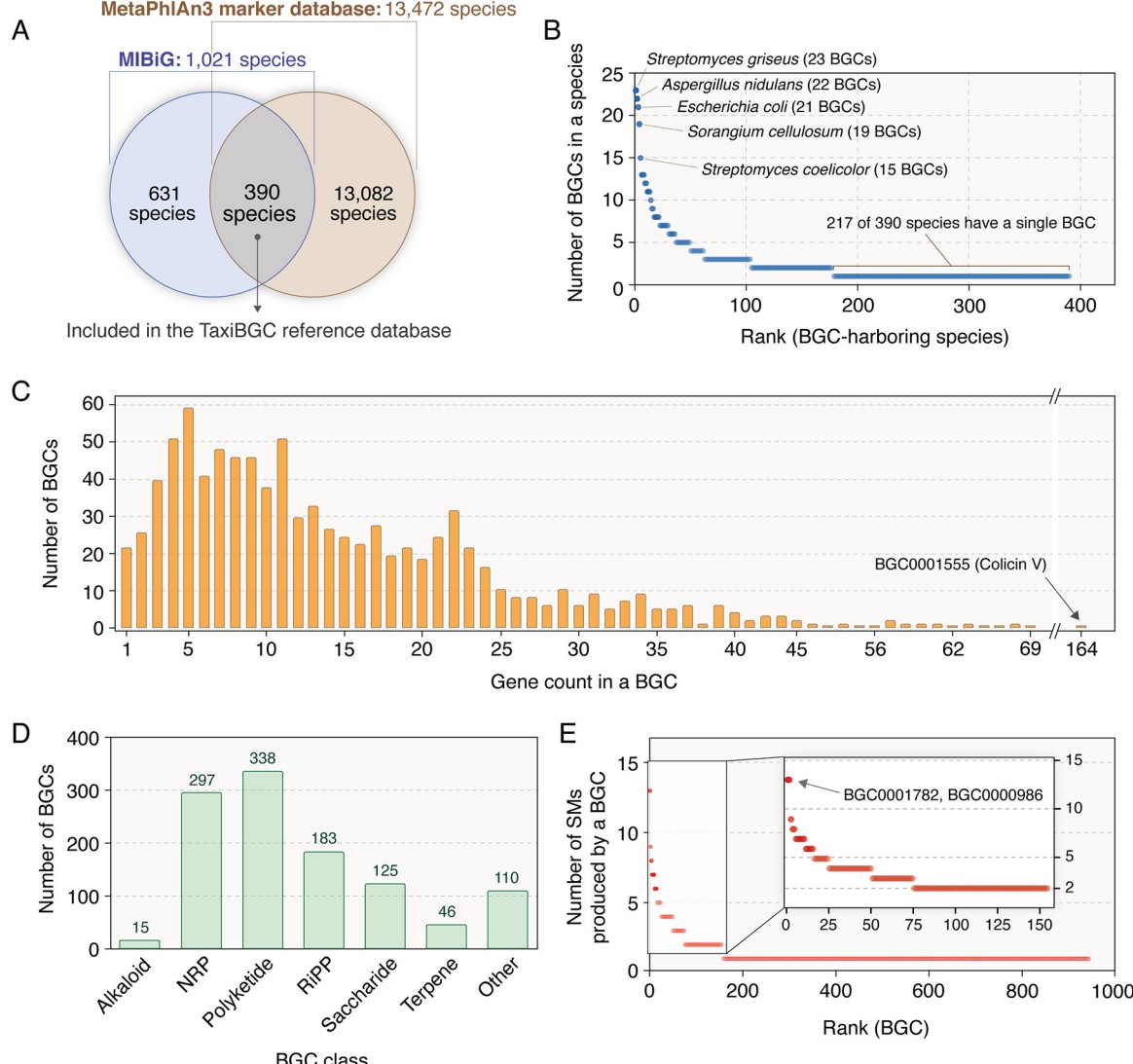

**FIG 2** Relationships among microbial species, BGCs, and SMs of the TaxiBGC reference database. The TaxiBGC reference database links 390 microbial species to their 953 BGCs, which encode for 1,154 unique SMs. (A) A total of 1,021 BGC-harboring species were obtained from the MIBiG database (as of January 2022). Among these species, 390 were common with those in the MetaPhlAn3 gene marker database (as of January 2022). (B) BGC-harboring microbial species rank ordered by their number of BGCs. (C) Number of BGCs with a given gene count. (D) Number of BGCs in each BGC class. Of note, a BGC can belong to multiple classes (Table S1). (E) BGCs rank ordered by how many SMs they produce.

on all 125 simulated metagenomes was selected as the optimal minimum BGC gene presence and BGC coverage cutoffs for TaxiBGC.

**Species, BGCs, and SMs of the TaxiBGC reference database.** The TaxiBGC reference database essentially provides information linking experimentally characterized BGCs and their corresponding SMs with their source organism (microbial species and/or strains). This database, which is relied on by the TaxiBGC pipeline during BGC prediction (Fig. 1A), is predefined and not assembled anew for each metagenome sample. For the construction of this database (Fig. 1B), we obtained annotated information regarding BGCs, SMs, and their respective source organisms from the MIBiG database (as of January 2022) (36). Since the first step of the TaxiBGC pipeline relies on MetaPhlAn3 for species-level taxonomic profiling, the TaxiBGC reference database takes into consideration only the 390 microbial species in MIBiG (along with their annotated BGC and SM information) found in common with those in the MetaPhlAn3 gene marker database (Fig. 2A).

In all, the TaxiBGC reference database encompasses 390 unique BGC-harboring microbial species, 953 unique experimentally characterized BGCs, and 1,169 total (1,154 unique) SMs. Of note, a BGC can be derived from multiple species, a BGC can encode multiple SMs, and more than one BGC can encode for the same SM. BGCs encoding nonribosomal peptides (NRPs) are the most prevalent among the 390 species (165 of 390 species, 42.3%); followed by polyketides (163 of 390 species, 41.8%); ribosomally synthesized and posttranslationally modified peptides (RiPPs) (117 of 390 species, 30.0%); others, i.e., not part of the six main categories provided in the MIBiG database and therefore collectively grouped into a single class (80 of 390 species, 20.5%); saccharides (74 of 390 species, 19.0%); terpenes (32 of 390 species, 8.2%); and alkaloids (12 of 390 species, 3.1%) (we followed the MIBiG standard notations for the major SM structure-based classes). BGCs, along with their classes/subclasses, and the taxa from which they are derived are described in detail in Table 1 and Table S1.

In the TaxiBGC reference database, the number of BGCs a species can harbor varies widely, ranging from 1 to as many as 21 (*Escherichia coli*), 22 (*Aspergillus nidulans*), or 23 (*Streptomyces griseus*); among all species, 217 (55.6% of 390) harbor only one BGC (Fig. 2B). The number of genes in a BGC also varies widely, ranging from a single gene to 164 genes (BGC0001555 encoding Colicin V); by and large, however, most of the 953 BGCs have less than 25 biosynthetic genes (Fig. 2C; see Table S2 in the supplemental material). Among the 953 BGCs, polyketides are the most abundant (338 of 953, 35.5%), followed by NRPs (297 of 953, 31.2%), RiPPs (183 of 953, 19.2%), saccharides (125 of 953, 13.1%), other (110 of 953, 11.5%), terpenes (46 of 953, 4.8%), and alkaloids (15 of 953, 1.6%) (Fig. 2D). Although both BGC0001782 and BGC000986 are known to encode a total of 13 different SMs (highest SM production potential among all 953 BGCs), the majority of BGCs produce only 1 SM (796 of 953 or 83.5%) (Fig. 2E).

**Factors determining the predictive performance of TaxiBGC.** In TaxiBGC, identifying whether a BGC is present (or not) in a metagenome is dependent mainly on the following two criteria: regarding a *BGC* gene being present, metagenomic reads should cover at least a minimum sequence length of that gene (i.e., BGC gene presence threshold); and concerning a *BGC* being present, at least a minimum proportion of its constituent genes should be present (i.e., BGC coverage threshold). On simulated metagenomes, of which the BGC compositions are known, TaxiBGC prediction performance was evaluated over a range of these thresholds (Materials and Methods). More specifically, a total of 400 pairs of thresholds for BGC gene presence (5 to 100% in increments of 5%) and BGC coverage (5 to 100% in increments of 5%) were tested on a total of 125 simulated metagenomes. The predicted BGCs were then compared with the actual BGCs to obtain prediction accuracy.

The best overall accuracy (mean $F_1$ score, 0.56) for predicting BGCs in all simulated metagenomes was achieved with a minimum BGC gene presence threshold of 5% and a BGC coverage threshold of 10% (Fig. 3A). The $F_1$ scores (the $F_1$ score is the harmonic mean of precision and recall) when calculated with these optimal thresholds (i.e., BGC gene presence, 5%; BGC coverage, 10%) were found to be very close to the $F_1$ scores found when using the pair of thresholds most accurate for each metagenome, which can vary sample to sample (Fig. 3B; see Table S3 in the supplemental material). This finding indicates that the chosen BGC gene presence (5%) and BGC coverage (10%) thresholds are suitable for the TaxiBGC pipeline for predicting BGCs in metagenomes. Thus, these thresholds were set as default parameters in TaxiBGC, although the user is given the option to change them in the command-line arguments based on the desired precision or recall.

We next investigated whether TaxiBGC accuracy is influenced by inherent characteristics of the simulated metagenomes, namely, mainly library size and relative abundances of BGC-harboring species. First, we found that $F_1$ scores for BGC prediction are generally not dependent on library size, although we did observe a slight decrease in $F_1$ scores with relatively shallow 5 million (paired-end) reads compared to those with 40 million and 80 million reads ($P = 0.047$, Mann-Whitney $U$ test) (Fig. 3C and Table S3).

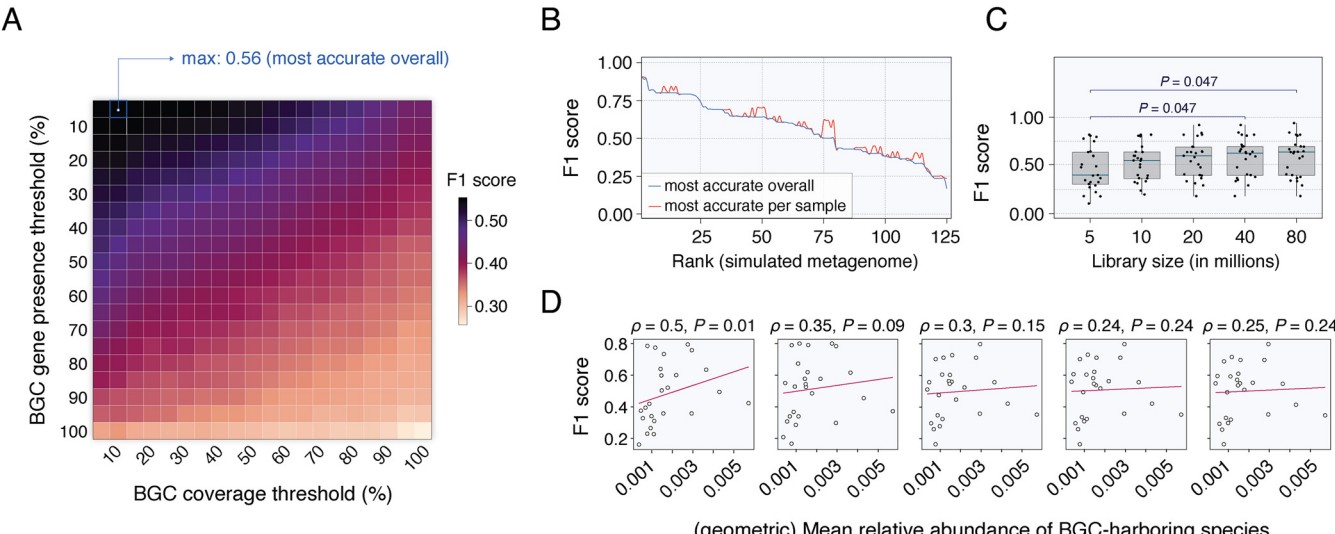

**FIG 3** TaxiBGC predictive performance is determined by the BGC gene presence and BGC coverage thresholds but not by metagenome library sizes or the mean relative abundances of BGC-harboring species. (A) TaxiBGC was used to predict BGCs in all 125 simulated metagenomes of mock microbial communities using 400 total pairwise combinations of BGC gene presence (5 to 100% in increments of 5%) and BGC coverage (5 to 100% in increments of 5%) thresholds. The best overall accuracy for predicting BGCs in simulated metagenomes was achieved with a minimum BGC gene presence and BGC coverage of 5% and 10%, respectively (mean $F_1$ score, 0.56). Therefore, these thresholds were set as defaults in TaxiBGC but can be modified by the user. (B) $F_1$ scores when using the overall optimal thresholds (blue line) resemble those when using the most accurate pair of thresholds for each metagenome sample (red line). (C) $F_1$ scores across various simulated metagenome library sizes when using TaxiBGC with the overall optimal thresholds. Prediction accuracy of TaxiBGC was reasonably consistent in simulated metagenomes with 10 million or more paired-end reads. Only the $F_1$ scores with a library size of 5 million reads were statistically different from those with library sizes of 40 million and 80 million reads ($P = 0.047$, Mann-Whitney $U$ test). Standard box-and-whisker plots (e.g., center line, median; box limits, upper and lower quartiles; whiskers, 1.5× interquartile range; points, outliers) are used to show distributions of $F_1$ scores. (D) No significant correlation was found between $F_1$ scores and (geometric) mean relative abundances of BGC-harboring species, except for in the case with 5 million reads (Spearman's $\rho = 0.5$, $P = 0.01$). Points in C and D correspond to individual simulated metagenomes. Scatter plots in D are for simulated metagenomes of 5, 10, 20, 40, and 80 million reads (from left to right).

Second, to see whether a higher relative abundance of BGC-harboring species is necessary for more favorable predictive performance, we next investigated the relationship between (geometric) mean relative abundance of those species in each sample and the resulting prediction accuracy (Fig. 3D; see Table S4 in the supplemental material). We found that $F_1$ scores did not correlate with the mean relative abundances of BGC-harboring species except in the case with 5 million reads. In summary, we recommend using TaxiBGC on metagenomes of at least 10 million reads, in which case the relative abundances of BGC-harboring species are not a significant contributing factor to prediction accuracy.

**Taxonomy-guided approach predicts BGCs more accurately than by directly detecting BGC genes.** As described above, the TaxiBGC pipeline predicts BGCs in metagenomes by first identifying the microbial species that harbor those BGCs. As an alternative to this taxonomy-guided approach, we can also predict BGCs more directly by mapping metagenomic reads onto known BGC genes (which is the basis of existing state-of-the-art techniques). To demonstrate the predictive capability of this direct BGC detection approach, we again tested a range of BGC gene presence and BGC coverage thresholds (5 to 100% with intervals of 5%) on the same 125 simulated metagenomes utilized above (Materials and Methods). From all possible pairwise combinations of thresholds, we found that a minimum BGC gene presence of 5% and a minimum BGC coverage of 90% led to the best mean $F_1$ score of 0.49 (see Fig. S1 in the supplemental material). Next, the predictive performance of the direct BGC detection approach was compared against that of TaxiBGC using the optimal thresholds found for each respective method. Having tested on simulated metagenomes, we found that TaxiBGC resulted in significantly higher $F_1$ scores ($P = 0.001$, Mann-Whitney $U$ test) (Fig. 4A). Moreover, positive predictive values (PPVs) were considerably higher for TaxiBGC ($P < 2 \times 10^{-16}$, Mann-Whitney $U$ test) (Fig. 4B), indicating that the taxonomy-guided method generally results in fewer false-positive predictions than true positives.

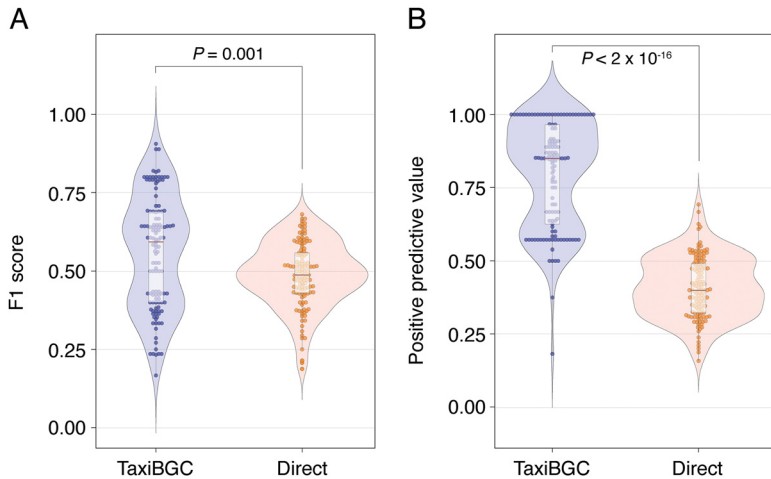

**FIG 4** TaxiBGC outperforms the direct BGC detection method in predicting BGCs in simulated metagenomes. Both $F_1$ scores (A) and positive predictive values (PPVs) (B) for BGC predictions in metagenomes were significantly higher when using TaxiBGC (mean $F_1$ score, 0.56; mean PPV, 0.80) than when directly detecting BGCs (mean $F_1$ score, 0.49; mean PPV, 0.41). The BGC gene presence (BGC coverage) thresholds for TaxiBGC and the direct BGC detection approach were set as 5% (10%) and 5% (90%), respectively, as these cutoffs led to the best overall prediction accuracies for all 125 simulated metagenomes. The center red line in the box-and-whisker plots denotes the median. Mann-Whitney $U$ test was used to test the level of significance.

**BGCs for bacteriocins are prevalent in microbiomes throughout the human body.** To demonstrate the utility of TaxiBGC, we applied our novel approach on 1,217 metagenome samples from the Human Microbiome Project (HMP-1-I and HMP-1-II) (37, 38). These metagenomes were sampled from the following five human body sites collected from healthy human individuals: gut, i.e., stool (472 samples); nose, i.e., anterior nares (116 samples); oral cavity (505 samples); skin, i.e., left/right retroauricular crease (26 samples); and vagina (98 samples) (detailed information regarding the metagenome samples used from these studies is available in Table S5 in the supplemental material; see Materials and Methods for sample exclusion criteria). We investigated each body site to determine the proportion of its samples predicted to have a certain BGC.

Across all five body sites, we identified a total of 24 unique BGCs present in at least one sample (Fig. 5 and Table S5). Fourteen of them were found at significantly different proportions in one or more body sites ($P < 0.05$, Fisher's exact test). Most (20 of 24) BGCs were exclusive to a particular body site (i.e., absent in metagenomes from other sites), while four BGCs encoding for an exopolysaccharide (BGC0000764), nisin A (BGC0000535), nisin O (BGC0001701), and ruminococcin A (BGC0000545) were found to be present in more than one body site.

The 24 BGCs mentioned above span four main BGC classes, as follows: 20 are RiPPs, one is an NRP, one is a saccharide, and two are from the 'other' class (Fig. 5). The 20 RiPPs, which include gassericins, lacticins, mutacins, nisins, and ruminococcin A, all belong to the bacteriocin group of antibiotics. Bacteriocins are a group of antimicrobial peptides that can kill or inhibit growth in closely related or nonrelated bacterial strains (39). Interestingly, there has recently been emerging evidence that supports bacteriocin-producing bacteria as probiotic strains for benefiting human health (40). In our study, mutacin K8 was found only in the oral microbiome ($P < 0.001$, Fisher's exact test) but only for 40 (or 7.9%) of the total 505 samples. Mutacin K8 is a major lantibiotic bacteriocin produced by *Streptococcus mutans*, which is a key bacterium for the formation of dental plaques. It has been shown to have potent activity against other streptococcal species in the oral cavity (41). Additionally, a BGC for nisin O (BGC0001701) was found in both skin and gut microbiomes ($P < 0.001$, Fisher's exact test); this BGC was predicted in 35 (or 7.4%) and 1 (or 3.9%) of the 472 gut microbiome and 26 skin microbiome samples, respectively. Lastly, BGCs for gassericin E (BGC0001388), gassericin S (BGC0001601), and gassericin T (BGC000619) were

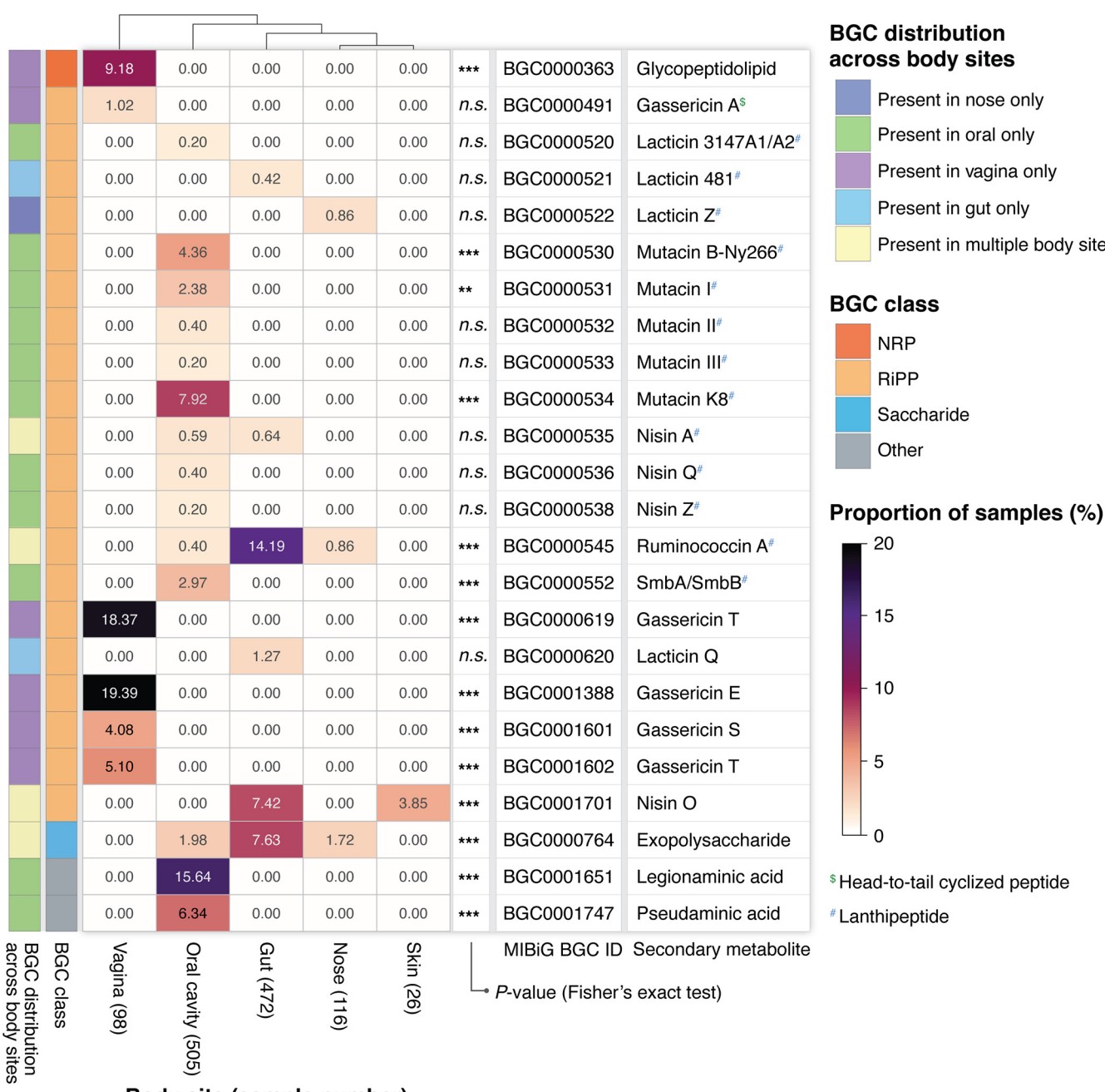

**FIG 5** Prevalence of BGCs in metagenomes sampled from five human body sites. TaxiBGC was used to profile metagenomic sequencing data from each body site (provided by the Human Microbiome Project) to determine the proportion of samples in which a BGC was predicted. A total of 24 BGCs were found to be present in at least one sample from all five body sites, of which 14 had significantly different proportions in one or more body sites ($P < 0.05$, Fisher's exact test). The color intensity in the heatmap reflects the proportion of samples (in a body site) wherein the corresponding BGC was found. Twenty BGCs were predicted in only one body site. The following symbols indicate the level of significance for the two-sided Fisher's exact test: $n.s.$, $P \geq 0.05$; **, $0.001 \leq P < 0.01$; and ***, $P < 0.001$. RiPP subclasses are denoted as follows: $^{\$}$, for head-to-tail cyclized peptide; and $^{\#}$, lanthipeptide.

predicted exclusively in the vaginal microbiome ($P < 0.001$, Fisher's exact test) for 19 (or 19.4%), 4 (or 4.1%), and 18 (or 18.4%) of the total 98 samples, respectively. A study showed that gassericin E, which was produced by *Lactobacillus gasseri* EV1461 isolated from the vagina of a healthy woman, inhibited the growth of pathogens associated with bacterial vaginosis (42). In all, our findings may motivate future studies on how bacteriocins govern ecological stability and dynamics within microbial niches of the human body.

**Gut microbial BGCs associate with human diseases.** We can also demonstrate TaxiBGC in the context of translational research to find BGCs associated with clinical

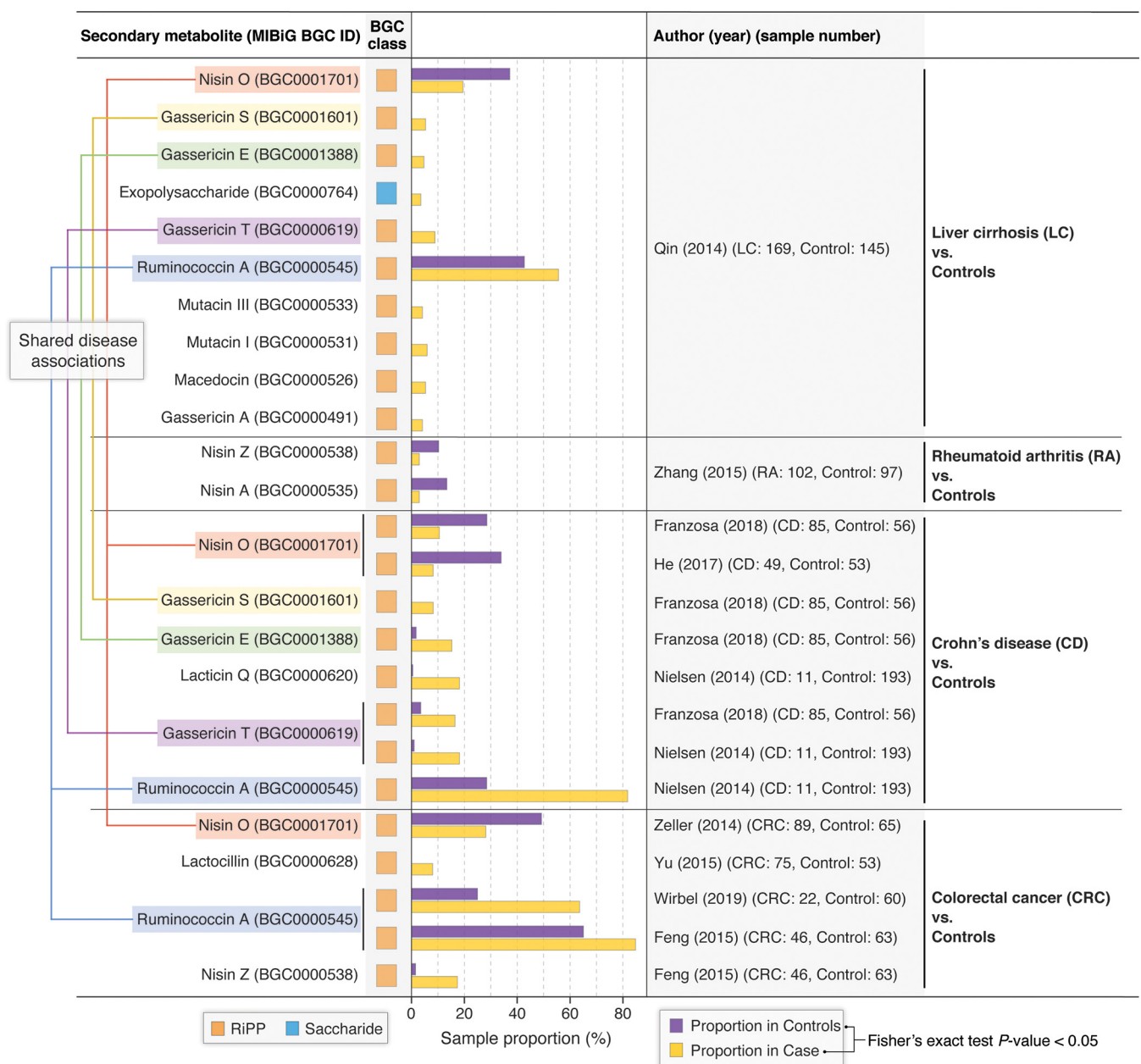

**FIG 6** TaxiBGC identifies gut microbial BGCs associated with several human pathologies. TaxiBGC was used to identify BGCs in gut microbiome samples from multiple case-control studies. In liver cirrhosis (LC), BGCs encoding 10 different SMs were found to be associated with either LC or control; in rheumatoid arthritis (RA), 2 BGCs encoding nisin A and nisin Z were found in lower proportions in RA than in controls; and in Crohn's disease (CD), BGCs encoding 5 different SMs (gassericin E, gassericin S, gassericin T, lacticin Q, and ruminococcin A) were found more often in CD. In contrast, a BGC for nisin O was found more frequently in controls; in colorectal cancer (CRC), three BGCs encoding lactocillin, nisin Z, and ruminococcin A were predicted in higher proportions in CRC. In comparison, one BGC encoding nisin O was found more frequently in controls. Case-control studies are designated with the lead author's last name and publication year. Sample numbers are shown in parentheses next to their respective cohort. Significance for association was calculated based on a two-sided Fisher's exact test ($P < 0.05$).

phenotypes of interest. We applied TaxiBGC on 1,433 gut microbiome (i.e., stool metagenome) samples from the following nine published case-control studies: three Crohn's disease (CD) studies (43–45), four colorectal cancer (CRC) studies (46–49), and one study each for rheumatoid arthritis (RA) (50) and liver cirrhosis (51) (detailed information regarding the metagenome samples used from these studies is available in Table S6 in the supplemental material; see Materials and Methods for sample exclusion criteria). As shown in Fig. 6, a total of 14 unique BGCs were found in gut microbiomes in significantly different

proportions between patients and control individuals ($P < 0.05$, Fisher's exact test), thereby associating BGCs and their SM production potential with human diseases.

In the study involving gut microbiomes from patients with LC and controls, ten BGCs were significantly different in proportions between LC patients and controls (Fig. 6), as follows: nine BGCs (encoding for an exopolysaccharide, gassericin A, gassericin E, gassericin S, gassericin T, mutacin I, mutacin III, macedocin, and ruminococcin A) were found to be present in a significantly higher proportion of LC patients than that in controls; on the other hand, only one BGC for nisin O was found more often in controls than in LC patients. Additionally, TaxiBGC found two BGCs encoding for nisin A and nisin Z less frequently in the gut microbiomes of patients with RA than in those of controls (Fig. 6).

Six BGCs were identified in significantly different proportions between patients with CD and controls (Fig. 6). Five BGCs encoding for gassericin E (in reference 43), gassericin S (in reference 43), gassericin T (in reference 43 and 44), lacticin Q (in reference 44), and ruminococcin A (in reference 44) were found more frequently in CD patients in separate individual studies. The one remaining BGC for nisin O (in reference 43 and 45) was predicted more often in controls; that these patterns for the nisin O-/gassericin T-encoding BGCs were observed in two different studies may indicate that our findings are robust.

Four BGCs were found to be associated with CRC (Fig. 6). Three gut microbial BGCs were found in significantly higher proportions in CRC (compared with controls); these BGCs encode lactocillin (in reference 48), nisin Z (in reference 46), and ruminococcin A (in reference 46 and 49). Alternatively, as was observed for CD studies, a BGC for nisin O was found in significantly higher proportions in controls than those in CRC patients (in reference 47). Among these four BGCs associated with CRC, three encode for bacteriocins (nisin O, nisin Z, and ruminococcin A), whereas one encodes for lactocillin, which is a thiopeptide antibiotic. Previously, BGCs for thiopeptides were found to be widely distributed in genomes of microbial species inhabiting the human gut, oral cavity, and vagina (16).

Interestingly, five of the disease-associated gut microbial BGCs mentioned above were shared across multiple gastrointestinal diseases. For example, BGC0001701 encoding nisin O was found more often in controls than in LC, CD, and CRC (see pink lines in Fig. 6). In contrast, in the same three diseases, BGC0000545 for ruminococcin A was found in higher proportions in cases than those in controls (see blue lines in Fig. 6). Lastly, three BGCs for gassericin E (BGC0001388; see green lines in Fig. 6), gassericin S (BGC0001601; see gold lines in Fig. 6), and gassericin T (BGC0000619; see purple lines in Fig. 6) were found in significantly higher proportions in both CD and LC patients than those in controls. Furthermore, all five of these BGCs encode bacteriocins. In summary, building upon previous findings of gut microbiome taxonomies or genes shared among different human diseases (52–54), we report for the first time experimentally characterized BGCs associated with the gut microbiome of multiple pathologies.

## DISCUSSION

Accurately predicting the biosynthetic potential of BGCs from microbiomes is currently one of the most important challenges in microbial natural product research. Here, we present TaxiBGC, a computational pipeline for predicting experimentally characterized, microbial BGCs with their known SMs in shotgun metagenomes. TaxiBGC employs the following three main steps for predicting BGCs: (i) identification of BGC-harboring species, (ii) initial prediction of BGCs based on the identified BGC-harboring species, and (iii) *in silico* confirmation of the predicted BGCs by mapping metagenomic reads onto the genes. We evaluated the prediction accuracy of TaxiBGC on simulated metagenomes (of mock microbial communities) constructed using varied combinations of species identities, species richness, and sequencing library sizes. Accordingly, we showed that BGC prediction using our taxonomy-guided approach (mean $F_1$ score,

0.56; and mean PPV score, 0.80) was superior to directly mapping reads onto BGC genes (mean $F_1$ score, 0.49; and mean PPV score, 0.41). Next, to identify BGCs (and thereby being able to estimate their corresponding SMs) associated with human health and disease, we demonstrated our TaxiBGC pipeline on metagenomes sampled from different body sites and on gut microbiomes collected from patients with liver cirrhosis, rheumatoid arthritis, Crohn's disease, or colorectal cancer. Our results show that human body sites harbor distinct microbial BGC signatures, which mostly encode bacteriocins. Furthermore, by characterizing the potential chemical milieu of the gut microbial environment for different diseases and their controls, our results provide a starting point for discovering stool-borne SM biomarkers and therapeutic leads.

Several limitations of TaxiBGC should be noted when interpreting our results. First, by design, BGC prediction by TaxiBGC relies on the successful identification of microbial species from metagenomes. In the case where MetaPhlAn3 (or other taxonomy profiling tools in future versions of TaxiBGC) cannot identify all the BGC-harboring species, TaxiBGC would not be able to accurately predict the entire breadth of BGCs. Second, predictions regarding the taxonomic origins of BGCs need to be interpreted with caution. This call for caution is because, in the "mixed bag of genes" context of metagenomes, gene elements of a particular BGC could originate from other BGCs and multiple species. Nevertheless, TaxiBGC can spark new and exciting hypotheses regarding the connections between microbiome BGCs and their source species, which can be validated in future empirical studies. Third, although the taxonomy-guided step in our approach can certainly reduce false-positive BGC predictions relative to true positives, the trade-off is that we may miss BGCs that are present in the metagenomes, thereby leading to type II errors (false negatives). Fourth, TaxiBGC provides only the chemical family—but not the specific molecular structure—for a small number of SM products, such as glycopeptidolipid and exopolysaccharide. This limitation is simply because more detailed information is not yet available in MIBiG. Fifth, TaxiBGC is designed to predict only known BGCs whose SMs have been verified by other research groups. Therefore, its predictions do not extend to putative (unvalidated) BGCs that have yet to be isolated and expressed in bacterial hosts. Finally, experimental verification of whether the BGCs (and their SMs) reported here are indeed implicated in health outcomes, e.g., either maintaining the structure of the so-called "healthy" gut microbiome (53, 55, 56) or eliciting immunopathogenic effects, remains an essential step but is beyond the scope of this computational study.

Despite the limitations mentioned above, TaxiBGC provides several advantages over currently available tools for BGC identification. First, TaxiBGC employs an assembly independent, read-based prediction method; this method solves a fundamental limitation of current BGC prediction tools that assume each BGC is encoded within a single contig in the genome or assembled metagenome, whereas in reality, genes of BGCs can be dispersed throughout several contigs. Second, TaxiBGC can be used to associate BGC presence with species abundance, which can help pinpoint which BGCs originate from which species under different phenotypic conditions. Third, TaxiBGC can simultaneously identify several classes of experimentally characterized BGCs in a metagenome sample. This comprehensive and unbiased detection is essential to deepen our understanding of the diversity of known BGCs and their SMs in a microbiome. Fourth, no parameter selection or optimization is necessary before running TaxiBGC. The only preinstalled software needed in TaxiBGC is MetaPhlAn3 for taxonomic species identification and BBMap v38.90 (or higher) for mapping metagenomic reads onto BGC genes.

In closing, we anticipate that TaxiBGC can facilitate the unbiased identification of experimentally characterized BGCs and their annotated SMs from shotgun metagenomes. It is also worth noting that TaxiBGC is not limited to the use in human microbiomes but can easily be applied to solve questions in other types of microbiomes (e.g., animal models, soil, seawater, and plants) for BGC identification. Additionally, we envision that our novel pipeline can contribute to our ongoing efforts to

computationally model the wide range of potential metabolite-driven interactions within complex microbial communities (57–59). Finally, to support these efforts and beyond, we plan to continuously expand and curate the current version of TaxiBGC with data newly added to MIBiG, as well as with updates to taxonomic profiling software.

## MATERIALS AND METHODS

**Constructing simulated metagenomes.** Simulated metagenomes were constructed according to the following steps. First, 25 metagenome samples were selected randomly from the Human Microbiome Project (HMP) (37, 38). Second, the species composition of each sample was determined using MetaPhlAn v3.0.13 (34). Third, mock microbial communities were made with BGC-harboring and non-BGC-harboring species while preserving the original species richness and relative abundance quantities found for each HMP metagenome (see Table S7 and Fig. S2 in the supplemental material). In other words, the actual species in each HMP sample were replaced with BGC-harboring species (from MIBiG) and non-BGC-harboring species (from NCBI GenBank) at random. This uneven distribution of species' counts and abundances allows the generation of simulated metagenomes as close to real metagenomes as possible. In all, 25 mock microbial communities, in which the unique species richness (i.e., count) ranged from ~40 to ~100, were used to construct simulated metagenomes (see Table S8 in the supplemental material). Fourth, for each species in a mock microbial community, the genome corresponding to its strain, or that of a randomly selected strain in case there were multiple strains for a particular species, was downloaded from NCBI GenBank. Fifth, the collection of genome sequences for all species of a mock community was converted into paired-end metagenomic reads of 100-bp read length using next-generation sequencing simulator for metagenomics (NeSSM) (60). In NeSSM, the number of reads generated from each genome is commensurate with the known community composition (thereby considering the names, total count, and relative abundances of the species) and the specified library size, resulting in a simulated metagenome for a given mock community. By various sequencing library sizes, a total of 125 simulated metagenomes [(25 unique sets of microbial species) × (5 library sizes: 5 million, 10 million, 20 million, 40 million, and 80 million paired-end reads)] were constructed.

**Predicting BGCs in simulated metagenomes using TaxiBGC.** First, species-level taxonomic profiling was performed on each simulated metagenome using MetaPhlAn v3.0.13 (34) with default parameters. Next, all identified species were queried in the TaxiBGC reference database for the first-pass prediction of BGCs that might be present. Then, as an *in silico* confirmation of those predicted BGCs, genes of those BGCs were searched for in the metagenomes using BBMap v38.90 (35) with minimum BGC gene presence (i.e., percentage length of a BGC gene covered by metagenomic reads) and BGC coverage (i.e., percentage of the total number of genes present for a particular BGC) cutoffs. BGCs that passed these two criteria were considered present in the given metagenome.

**Predicting BGCs in simulated metagenomes through direct detection of BGC genes.** The direct BGC gene detection strategy was used to predict BGCs in a simulated metagenome without taxonomic profiling. Accounting for all BGCs in the MIBiG database (1,905 BGCs as of January 2022) (Table S1), this approach searches for BGC genes in simulated metagenomes using BBMap v38.90 with a pair of minimum BGC gene presence and BGC coverage cutoffs. BGCs that passed these two criteria were considered present in the given metagenome.

**$F_1$ scores for evaluating TaxiBGC prediction accuracy.** The $F_1$ score, defined as the harmonic mean of precision and recall, is calculated as follows:

$$F_1 = \frac{2}{\frac{1}{precision} + \frac{1}{recall}}$$

where

$$precision = \frac{TP}{TP + FP}$$

and

$$recall = \frac{TP}{TP + FN}$$

Here, for a given metagenome, *TP* (true positives) is the number of BGCs that were correctly predicted to be present; *FP* (false positives) is the number of BGCs that were incorrectly predicted to be present; and *FN* (false negatives) is the number of BGCs that were missed, i.e., incorrectly predicted to be absent by TaxiBGC.

**Exclusion criteria for metagenomic data sets used in TaxiBGC demonstration.** Metagenome samples from the following were excluded from the current study: (i) subjects undergoing dietary/medication interventions, (ii) metagenome samples with less than 5 million reads, (iii) metagenome samples sequenced as single-end reads, and (iv) samples collected from patients with comorbidities. In the case where multiple runs were available for a sample, only the sequencing run with the highest read count (of at least 5 million reads) was included for analysis. If multiple samples were collected from the same

individual across different time points, then only the sample collected at the first recorded time point was considered. In all, the TaxiBGC pipeline was applied on 2,650 publicly available, human-derived metagenomes from 11 independent published studies. Raw sequence files of all metagenomes were downloaded from the NCBI Sequence Read Archive.

**Data availability.** TaxiBGC is an open-source command-line tool that is implemented in Python and bash. To avoid dependency conflicts, TaxiBGC can be installed via Anaconda (https://anaconda.org/danielchang2002/TaxiBGC). The source code, MIBiG BGC gene sequences, the TaxiBGC reference database, and complete instructions for installation and usage are freely available online at https://github.com/danielchang2002/TaxiBGC_2022.

## SUPPLEMENTAL MATERIAL

Supplemental material is available online only.

**FIG S1**, TIF file, 1.3 MB.
**FIG S2**, TIF file, 1.7 MB.
**TABLE S1**, XLSX file, 0.1 MB.
**TABLE S2**, XLSX file, 0.03 MB.
**TABLE S3**, XLSX file, 0.01 MB.
**TABLE S4**, XLSX file, 0.01 MB.
**TABLE S5**, XLSX file, 0.1 MB.
**TABLE S6**, XLSX file, 0.2 MB.
**TABLE S7**, XLSX file, 0.1 MB.
**TABLE S8**, XLSX file, 0.01 MB.

## ACKNOWLEDGMENTS

We declare no conflict of interest.

We thank David N. Rider and Ronald L. Neuharth for their help in downloading all metagenome samples used in our study.

This work was supported in part by the Mayo Clinic Center for Individualized Medicine (J.S.) and Mark E. and Mary A. Davis to Mayo Clinic Center for Individualized Medicine (J.M.D. and J.S.). The funders had no role in the study design, data collection and interpretation, or the decision to submit the work for publication.

J.S. conceived the problem. V.K.G., U.B., and J.S. designed all analytical methodologies. V.K.G., U.B., and D.C. built the TaxiBGC computational pipeline. V.K.G. performed all analyses. All authors analyzed the data. V.K.G., U.B., and J.S. wrote the manuscript, with editorial contributions from other authors.

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
