## [Reviewer comments · mSystems]

TaxiBGC: a Taxonomy-guided Approach for Profiling Experimentally Characterized Microbial Biosynthetic Gene Clusters and Secondary Metabolite Production Potential in Metagenomes

Vinod Gupta, Utpal Bakshi, Daniel Chang, Aileen Lee, John Davis, Sriram Chandrasekaran, Yong-Su Jin, Michael Freeman, and Jaeyun Sung

Corresponding Author(s): Jaeyun Sung, Mayo Clinic

Review Timeline:

Submission Date:

September 22, 2022

Accepted:

October 14, 2022

Editor: Nicola Segata

Reviewer(s): The reviewers have opted to remain anonymous.

Transaction Report:

DOI: <https://doi.org/10.1128/msystems.00925-22>

October 14, 2022

Dr. Jaeyun Sung
Mayo Clinic
Rochester, MN 55905

Re: mSystems00925-22 (TaxiBGC: a Taxonomy-guided Approach for Profiling Experimentally Characterized Microbial Biosynthetic Gene Clusters and Secondary Metabolite Production Potential in Metagenomes)

Dear Dr. Jaeyun Sung: Your manuscript has been accepted, and I am forwarding it to the ASM Journals Department for publication. For your reference, ASM Journals' address is given below. Before it can be scheduled for publication, your manuscript will be checked by the mSystems production staff to make sure that all elements meet the technical requirements for publication. They will contact you if anything needs to be revised before copyediting and production can begin. Otherwise, you will be notified when your proofs are ready to be viewed.

Publication Fees:

If you would like to submit a potential Featured Image, please email a file and a short legend to mSystems@asmusa.org. Please note that we can only consider images that (i) the authors created or own and (ii) have not been previously published. By submitting, you agree that the image can be used under the same terms as the published article. File requirements: square dimensions (4" x 4"), 300 dpi resolution, RGB colorspace, TIF file format.

We recognize that the video files can become quite large, and so to avoid quality loss ASM suggests sending the video file via <https://www.wetransfer.com/>. When you have a final version of the video and the still ready to share, please send it to mSystems staff at mSystems@asmusa.org.

Sincerely,

Nicola Segata
Editor, mSystems

Journals Department
Figure S1: Accept
Table S1: Accept
Table S5: Accept
Table S7: Accept
Figure S2: Accept
Table S4: Accept
Table S6: Accept
Table S8: Accept
Table S3: Accept
Table S2: Accept